# Mood Lifters for Graduate Students and Young Adults: A Mixed-Methods Investigation into Mechanisms of Change in Online Group Therapy

**DOI:** 10.3390/bs14030252

**Published:** 2024-03-20

**Authors:** Elena L. Pokowitz, Neema Prakash, Dennis Planaj, Sophia Oprandi, Patricia J. Deldin

**Affiliations:** Department of Psychology, University of Michigan, 530 Church Street, Ann Arbor, MI 48109, USA; neemapra@umich.edu (N.P.); dplanaj@umich.edu (D.P.); oprandi@umich.edu (S.O.); pjdeldin@umich.edu (P.J.D.)

**Keywords:** qualitative, mixed methods, mechanisms, mediators, treatment, group therapy

## Abstract

Mood Lifters (ML) is a dimensional, group-based, peer-led mental health program that has shown efficacy in mitigating psychopathology and promoting wellness within a variety of populations. There is not yet evidence for mechanism(s) driving these changes. Qualitative data exploring the lived experiences of participants may be a unique way to develop hypotheses about the potential mechanisms driving change. The current study used qualitative and quantitative data from the ML for Graduate Students (ML-GS) and Young Adults (ML-YA) studies to generate hypotheses regarding potential mechanisms of changes experienced in ML. Seventy-nine graduate students and fifty-nine non-student young adults provided quantitative and qualitative feedback after participating in a 12-week virtual ML program. Inductive qualitative analysis was conducted within a reflexive thematic framework. Descriptive statistics of quantitative measures were also calculated. Three themes suggest potential directions for future mechanism research on virtual ML programs. Most participants emphasized the social connections built in groups as the most potent and powerful aspects of ML, while others focused on the design or content of the program. Quantitative data presented contextualize the lived experiences of participants. Future ML research should explore the three themes identified within this study through continued qualitative and quantitative data collection and analysis. NCT05078450.

## 1. Introduction

Mood Lifters (ML), a biopsychosocial, transdiagnostic treatment model, was developed in response to the growing conceptualization of psychopathology as mechanistically (i.e., RDoC, [1]) and functionally (i.e., HiTOP, [2]) dimensional in nature. ML is a peer-led group therapy program in which individuals are introduced to content and skills across biopsychosocial domains, including sleep, body, mind, mood, behavior, and social content. Trained peer leaders facilitate community building and discussion amongst participants, guiding them through the material and encouraging skill practice between sessions in the form of “earning points”. Participants meet once a week for 12 weeks, completing one module each week over the course of an hour. ML has been conceptualized as a primary source of care for some and an adjunctive source of care for others. For individuals experiencing no, mild, or moderate symptoms, ML may function as prevention and/or intervention without the need for other mental health care (e.g., [3]). For those experiencing more severe symptom levels, ML may be adjunctive to other individual or group-based mental health care (e.g., [4]). For detailed information on the ML program and content, see Pokowitz et al., 2024 [5]. The efficacy and effectiveness of ML have previously been established across various populations, including the general adult population [6,7,8], seniors [9], individuals with bipolar disorder [4], parents of children with complex medical needs [10], and student athletes [11]. Previous research has also shown that peer leaders are equally effective in delivering the ML program as licensed clinicians while also conferring other benefits, such as lowering mental health stigma, creating an informal community space, and lowering the cost of access to care [7]. Most recently, ML was tested in a randomized control trial of graduate students [3] and non-student young adults. Though ML continues to show positive impacts on psychopathology and mental wellness across study types and populations, there is not yet evidence for a mechanism or combination of mechanisms driving these changes.

Treatment research has progressed tremendously throughout the 20th and 21st centuries, driving not only improvements in our quality of care but also spurring investigations into the mediators and mechanisms that are underlying the changes we see across various therapeutic modalities. Studying these mechanisms allows both researchers and clinicians to further optimize care, allowing for more effective implementation strategies, better discernment of who might benefit the most from different treatment types, and a clearer understanding of what types of outcomes clients might experience within those treatment types [12]. One potentially underutilized method of mechanism exploration is the analysis of qualitative data. Qualitative data provides a unique lens into the lived experiences of clients in a treatment program, allowing researchers to explore the salient processes, experiences, and changes experienced in therapy from the perspective of the individual engaging in the therapeutic process (e.g., [13,14]).

Mechanisms of change in group therapy have been previously examined across other approaches to care. Some research on cognitive behavioral group therapy (CBGT), for example, has focused on skills-based mediators, including evidence for changes in cognitive reappraisal, self-focused attention, and anticipatory and post-event processing as primary mechanisms driving symptom changes (e.g., [15,16]). Other research on CBGT has focused on the therapeutic alliance, finding that symptom changes over the course of therapy were mediated by a working alliance between the participants and the therapist at various points of the program [17]. More general research has looked to define common factors across group therapeutic environments. Yalom [18] details eleven factors of group therapy, including universalism, altruism, cohesion, catharsis, imparting information, imitation and modeling, instilling hope, developing social skills, learning interpersonal skills, and support. These factors have more recently been conceptualized as a potential single, higher-order factor of group therapy that can be measured with fewer and more general self-report questions [19].

## 2. Objectives

The current study sought to understand participants’ experiences in virtual Mood Lifters for Graduate Students (ML-GS) and Mood Lifters for Young Adults (ML-YA) programs between 2021 and 2022 using qualitative and quantitative feedback measures. The current study’s results will guide the investigation of potential key mechanisms of Mood Lifters and inform future iterations of the ML-GS and ML-YA programs specifically as well as group-based, transdiagnostic programs more generally.

## 3. Methods

### 3.1. Participants

Participants were eligible to join the study if they were graduate students or non-student young adults between the ages of 22 and 34 and were not experiencing acute suicidality, mania, and/or psychosis. Suicidality was assessed using the Patient Health Questionnaire-9 (PHQ-9, [20]) and study-specific follow-up questions assessed upon endorsement. Potential psychosis was assessed using the Community Assessment of Psychotic Experiences (CAPE, [21]). Risk for mania was assessed using the Mood Disorder Questionnaire (MDQ, [22]). Upon endorsing risk for suicidality, mania, and/or psychosis, potential participants were contacted via phone and briefly interviewed before eligibility was determined. Graduate students were assigned to ML-GS groups and non-student young adults were assigned to ML-YA groups. After receiving consent and assessing eligibility, 227 participants (129 graduate students and 98 young adults) were randomly assigned to the intervention or took part in the intervention after participating in the waitlist control group. A total of 31 (13.7%) participants never attended a meeting (17 graduate students, 14 young adults), and 26 (11.5%) participants dropped out of their group after attending at least one meeting (13 graduate students, 13 young adults). Participants were not required to have a prior mental illness diagnosis or experience. For the current study, the analysis sample included 79 ML-GS participants and 59 ML-YA participants who completed the feedback surveys embedded within the post-treatment assessment. See Table 1 for demographic information.

### 3.2. Procedures

Mood Lifters: ML-GS and ML-YA are 12-week ML programs specifically adapted for the graduate student and young adult populations, respectively. Program content consisted of topics within the six core wellness domains of ML (sleep, body, behavior, mood, mind, social), with some topics created specifically for the ML-GS and ML-YA adaptations (e.g., imposter syndrome, advisor/advisee relationships, boss/employee relationship). Groups were made up of 10–15 individuals self-identifying as either a graduate or a young adult. Hour-long sessions occurred weekly over HIPAA-compliant Zoom software for the 12 weeks of the program. Peer leaders were graduate students from the University of Michigan or non-student young adults from the surrounding metro area who had previously participated in a ML program themselves. Leaders took part in an eight-hour online training program, in vivo training during their first group, as well as weekly supervision with a clinical psychologist (PJD).

Feedback surveys: At the end of the 12-week intervention period, all participants were given the opportunity to provide feedback regarding their experience with Mood Lifters. Participants completed their feedback surveys within one week of the final meeting. Quantitative (e.g., “On a scale of 1 to 10, how useful was the Mood Lifters program to you?”) as well as qualitative (e.g., “What was the most powerful aspect of the group?”) questions were presented. The complete post-participation feedback survey is available in Appendix A. Follow-up (1-month and 6-month) surveys were collected to measure potentially sustained improvements or changes in psychological wellbeing over time. Qualitative feedback regarding program experiences was not collected at either follow-up timepoint.

Therapeutic factor inventory-8 (TFI-8): The TFI-8, initially developed by Lese and MacNair-Semands [23] and later shortened to eight items by [19], was originally designed to measure the eleven factors of group psychotherapy as described by Yalom [18]. The eight-item version used in the current study is both valid and reliable in measuring a more overarching conceptualization of the mechanism of group therapy [19]. Total scale scores range from 8 to 56, with higher scores reflecting a more cohesive group environment conducive to the group therapy process.

Group Climate Questionnaire—Engagement Subscale: The Group Climate Questionnaire (GCQ) measures how group members perceive various factors associated with a group therapy environment [24]. The engagement subscale, collected in the current study, specifically measures positive and constructive therapeutic work within a group, including understanding, cohesion, confrontation, and self-disclosures [25]. Total subscale scores reflect the average of the five item responses and range from 0 to 6, with higher scores reflecting perceptions of a more engaged group overall. The GCQ is widely used in group therapy settings and is valid and reliable [25].

Attendance: Participant attendance at Mood Lifters sessions was tracked each week by peer leaders.

Ethical approval: The University of Michigan Institutional Review Board granted study approval (HUM00163570). All participants gave informed consent.

### 3.3. Analysis

Qualitative analysis was conducted in NVivo and Microsoft Excel softwares. The current study used an inductive approach to coding within a reflexive thematic framework [26,27]. The coders approached data through an essentialist epistemological stance, such that participant responses were treated as accurately reflecting the lived experiences of individuals in the Mood Lifters program. Coding was conducted by a team of four, including two clinical science doctoral students (EP, NP) and two undergraduate research assistants (SO, DP). All participant feedback was double-coded. Any discrepancies were resolved through team discussion and consensus. The codebook was iterative in nature and updated throughout the coding process.

After finalizing the codebook, the coding team proposed three overarching themes that covered the breadth and depth of the available qualitative data. Each member of the coding team independently proposed which codes they deemed a good fit within each of the three themes. Consensus was reached through team discussion. Participant quotes were then used to name each of the themes. Descriptive attendance statistics, quantitative feedback prompts, TFI-8 scores, and GCQ-E scores were calculated to contextualize the qualitative feedback analyzed in the current study and better understand participants’ experiences in the program [28,29].

## 4. Results

### 4.1. Quantitative Measures

On a scale from 1 to 10, participants rated Mood Lifters’ usefulness at an average score of 7.2 (SD = 2.1). 35.5% of the respondents (49 of 138) reported wanting to repeat the program, and 80.4% (111 of 138) expressed interest in being contacted when new modules are developed. A total of 84.8% (117 of 138) of participants somewhat or strongly agreed that Mood Lifters enabled them to manage their mental health better (rated on a scale of 1 to 5, average = 4.1). 30.4% (42 of 138) were interested in becoming peer leaders for the program. See Table 2 for descriptive statistics of all quantitative feedback questions.

Participants had an average TFI-8 score of 37.5 (SD = 8.6) and an average GCQ-E score of 3.2 (SD = 0.4). Both are in the mid-range of potential scores on each measure. Participants attended an average of 10.6 out of the 12 meetings offered (SD = 2.0, ranging from 1 to 12).

### 4.2. The Most Powerful Aspects of Mood Lifters

Participants described the most powerful aspects of the Mood Lifters program across three overarching qualitative themes (see Figure 1). Each theme is related to a core way in which the program was designed to offer mental health care.

*“Structure and guidance.”*: Participants reported that some of Mood Lifters’ most potent aspects are how it is designed to support positive change in mental wellness. Overall, approximately 22% of the feedback collected (24% of young adults and 20% of graduate students) mentioned some characteristic of the structure or infrastructure of the program as one of the most potent aspects of Mood Lifters. These individuals attended an average of 10.8 meetings (ranging from 7 to 12) and rated the program’s usefulness at an average of 7.1 out of 10 (ranging from 3 to 10). They had an average TFI-8 score of 35.5 (range from 8 to 53) and an average GCQ-E score of 3.0 (range from 1.6 to 5). This theme emphasizes the value of the program’s design, including peer leaders’ power in guiding participants through the material.

*“...accepting and supportive facilitators.”* (105w, young adult)

*“Having strong and compassionate group leaders.”* (173w, young adult)

*“Having it led by a student like us.”* (32, graduate student)

A consistent, weekly space reserved for wellness felt validating and encouraged consideration of mental health.

*“It felt like a very safe space to work on these things.”* (124, young adult)

*“Having dedicated time set aside to work on my wellness.”* (240w, young adult)

Additional core aspects of the program’s unique design, including the gamified homework program (point system) and the breadth of skills offerings, were mentioned.

*“The point systems as a way to keep track of positive changes we were making.”* (121, graduate student)

*“Knowing that I could take what was useful and ‘leave’ what was not.”* (88, young adult)

*“Creating a community.”*: Most participants mentioned the value of other people in their feedback, with about 67% (68% of young adults and 66% of graduate students) speaking to the Mood Lifters community as one of the program’s most powerful aspects. Participants who brought up this theme attended an average of 10.6 meetings (ranging from 4 to 12) and rated the program’s usefulness at an average of 7.4 out of 10 (ranging from 1 to 10). They additionally had an average TFI-8 score of 38.8 (range from 16 to 56) and an average GCQ-E score of 3.5 (range from 0.6 to 6). Hearing from others with shared experiences validated and broadened the perspectives of many participants.

*“Hearing different takes on similar problems.”* (182, graduate student)

*“Having what can be lonely experiences be affirmed by others who have experienced similar things.”* (115, graduate student)

*“To hear from others the victories and/or struggles. It give[s] a sense that you are not alone and was a good source of encouragement.”* (332w, young adult)

Others spoke of the importance of having a community space to share their feelings and perspectives:

*“Having a group that I felt comfortable with sharing things.”* (109w, young adult)

*“I like that we all get to share our thoughts and help each other.”* (313, graduate student)

Bringing together so many different types of people worked to create a community, giving graduate students and young adults a space to discuss their mental wellness with others who understood what they were going through.

*“Sharing a space with people I wouldn’t normally interact with on such a personal level, and finding out that we all have so much in common.”* (72w, young adult)

*“Being able to connect and feel affirmed by other students going through similar experiences.”* (46w, graduate student)

*“Being able to talk with people going through the same things as me, realizing that many of us share the same concerns, fears, and negative thought patterns.”* (335, graduate student)

*“It was easier to find a sense of community and belonging, when in reality we often feel so isolated and that our problems are uniquely our own.”* (242w, young adult)

*“A toolkit to leverage in times of need.”*: Mood Lifters works to provide individuals with the tools for positive change, offering a wide variety of evidence-based strategies that can be used to work toward mental wellness. Approximately 29% of participants (25% of young adults and 32% of graduate students) shared that these tools were one of the most powerful aspects of the program. These individuals attended an average of 10.5 meetings (ranging from 1 to 12) and rated the program’s usefulness at an average of 6.9 out of 10 (ranging from 3 to 10). They also had an average TFI-8 score of 36.4 (ranging from 8 to 50) and an average GCQ-E score of 3.1 (ranging from 1 to 5.2). Mood Lifters aims to change thoughts, behaviors, and emotions across the duration of the program, setting individuals up for success in managing future stressors. Some participants offered specific content that they found the most powerful:

*“Learning about thinking traps and behaviors and how to manage them was really important to me.”* (283, graduate student)

*“I really appreciated learning about the theory and research around common things I had struggled with before (for example, ways of dealing with negative emotions). There were several times where I had “aha!” moments, realizing that I wasn’t alone, this was a normal thing to experience, and that there were strategies to effectively deal with issues that popped up. Naming something is the first step to changing it, and I am very grateful to have received the names and language for common issues in mental/emotional life.”* (297w, graduate student)

*“I loved the Imposter Syndrome section.”* (353, young adult)

*“It gave me some more language to use when thinking about my problems, or gave me frameworks in which to think about my problems.”* (214w, graduate student)

Others provided personal testimonials about the importance of the changes they experienced during the program in describing its most potent aspects.

*“I’ve definitely gotten better at preventing myself from getting caught in thinking traps and I have some strategies I can try to implement to improve my mood.”* (283, graduate student)

*“Feeling like I had control and ownership over my mood and health.”* (68, graduate student)

*“Figuring out where my self-confidence lies and how best to harness it.”* (317, young adult)

## 5. Discussion

The Mood Lifters for Graduate Students and Mood Lifters for Young Adults programs seek to support young adults in navigating transdiagnostic mental health and wellness throughout a transitional time in their lives. Previous studies have shown that ML is efficacious and effective in reducing symptoms of psychopathology and promoting mental wellness across various populations and adaptations (e.g., [3,4,7,8,9,10,11]), yet the mechanism driving these changes has yet to be determined. The current study sought to explore potential mechanisms of change by analyzing participants’ lived experiences of engagement with the ML program. In providing post-treatment feedback on their program experiences, participants described the most powerful aspects of ML across three distinct but related themes: (1) “structure and guidance,” (2) “creating a community,” and (3) “a toolkit to leverage in times of need.” These themes were contextualized within descriptive statistics reflecting participants’ meeting attendance, program ratings, and measures of perceived group therapeutic factors—a mixed-methods approach to understanding participant experiences, allowing for hypothesis generation that will guide future ML research.

“Structure and guidance” reflects the opinions of a subset of participants who felt that the program’s design was the most powerful aspect of their ML experience. This included participants who emphasized the program’s use of peer leaders, the points system, the individual feedback sessions, the consistency and structure of the groups, the use of unconditional positive regard by leaders, and the breadth of content offered. In comparison, those who mentioned “a toolkit to leverage in times of need” specifically emphasized the content itself (rather than its breadth) and how that content prompted personal considerations of mental health and positive changes over time. “Creating a community” reflects the feelings of most participants who provided feedback, highlighting the importance of the group setting and how sharing, listening, and discussing with others allowed for validation and expansion of understanding. While these three themes reflect distinct participant experiences, all three are related. For example, the group setting that encourages community among participants is inherently part of the program’s design, while content discussion depends on that community and its dynamics. It is important to note that all participants had access to the same base content within the same program infrastructure and a group therapeutic environment. Individual differences in how the program was perceived may provide insight into why and how ML promotes mental wellness and mitigates psychopathology symptoms.

The use of teleconferencing software to deliver the ML program may have impacted the qualitative and quantitative feedback shared by participants, as there is some evidence that the use of telemedicine may affect group therapeutic factors [30,31]. While certain factors of group therapy such as universality, altruism, catharsis, imparting of information, support, and instilling hope may be equally achievable in person and in teletherapy formats, other factors such as cohesion, imitation and modeling, developing social skills, and learning interpersonal skills may be more difficult to achieve in a virtual ML group. Group therapeutic factors and participants’ reported experiences of the ML program may have been similarly altered by the use of peer leaders in place of a licensed clinician, though previous research has shown that peer leaders are equally as effective in promoting positive changes in mental health and wellness [7]. Future research exploring the mechanisms of change experienced in the ML program should analyze data from both in-person and virtual groups in order to determine whether differences in group cohesion are present across delivery types for this specific therapeutic approach.

Though the average number of meetings attended was similar across the thematically grouped subsets of participants, those who mentioned the two logistics-related themes tended towards lower usefulness ratings of the program as a whole. Individuals in those two subgroups similarly trended towards lower TFI-8 and GCQ-E scores than those who emphasized their connection with other participants. These findings suggest that individuals who most benefit from ML may find the program’s group aspect to be one of the most salient features. Alternatively, individuals who rate the group therapeutic aspects higher than their peers may have had a better personal experience within the group environment than those who tend to highlight the logistical side of the program instead. Regardless of potential directionality, constructs surrounding social support and interpersonal connection should be explored in future ML studies as potential mechanisms driving the changes observed in psychopathology throughout the program. Due to the presence of distinct themes within this analysis, it is additionally possible that ML works differently for different individuals; while social support and connection may be the driver for some, more direct skill learning may be the driver for others. In this sense, ML may provide many different paths toward mental wellness, as theorized in Pokowitz et al., 2024 [5].

Previous research has highlighted various aspects of the therapeutic process as key mechanisms driving change, including skills gained (e.g., [15,16]), common group therapeutic factors (e.g., [18,19]), and the therapeutic alliance between participants and group leaders (e.g., [17]). The current study highlights multiple areas aligning with these key and established aspects of group therapy (e.g., skills gained, importance of peer leaders, group cohesion) while seeming to further underscore some higher-order factor representing the importance of interpersonal connection and understanding. In this way, the current study pushes the boundaries of the currently understood common factors of group therapy towards a more general construct reflecting the feeling that “I am not alone”. Further qualitative and quantitative research should be conducted within Mood Lifters and other group therapeutic approaches to better understand this construct and how it might drive the changes that participants experience during the program.

Other quantitative feedback measures speak indirectly to the program’s acceptability within the young adult population. Just over a third of participants wanted to repeat the program, and over 80% requested access to new modules upon their development. Approximately 30% were interested in becoming peer leaders, reflecting the continued success of an integral feature of the program’s self-sustaining design. Responses to other feedback prompts were generally positive but spread across the rating scales (see Table 2), suggesting that Mood Lifters as a mental wellness program is acceptable and valuable to many, but not all, within this population. Future studies should explore the mechanisms and benefits of ML across individual differences in order to provide more effective care to those who need it.

The current study must be considered within its limitations when applying its findings to future studies of ML and other group-based, transdiagnostic programs. The qualitative and quantitative analyses within this study were limited to participants who filled out the feedback survey, potentially skewing results toward those who wanted to give feedback or were more likely to fill out the post-treatment survey in general. Further, these results are limited to young adults and graduate students between the ages of 22 and 33 with somewhat homogeneous demographics (i.e., primarily white and female), thus potentially limiting the generalization of these results to other ML adaptations or more diverse populations. Future research on ML should better incentivize feedback surveys, including exit surveys intended for those who drop out of the program. Researchers should additionally work to recruit more diverse populations and larger study samples through partnerships with community organizations outside the university system. Limitations also include the reliance on self-report data, which often confers concerns with validity, and that the feedback surveys were not reassessed during 1-month and 6-month follow-up periods. The use of self-report data in the current study, however, should be conceptualized as both a limitation and a strength, as understanding the subjective experiences of participants in the ML program may provide a richer understanding of potential mechanisms than objective quantitative data could provide.

## Figures and Tables

**Figure 1 behavsci-14-00252-f001:**
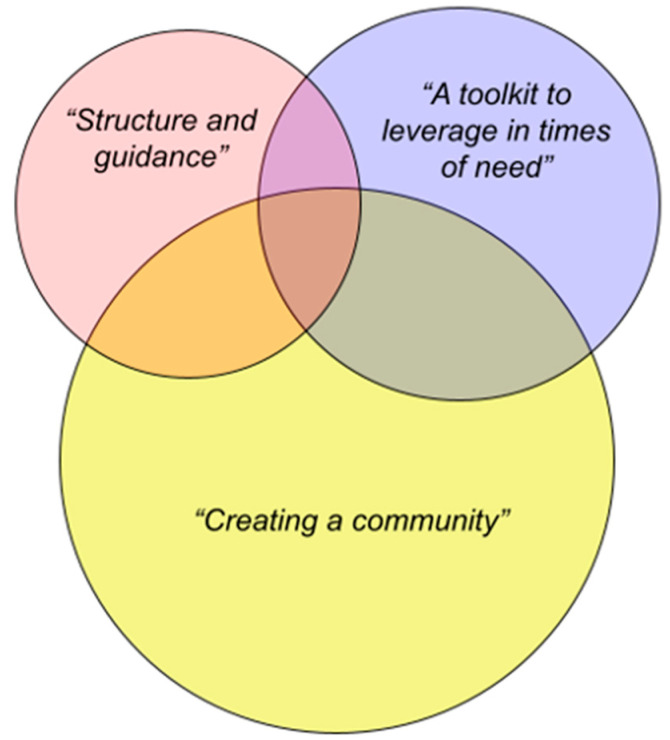
Thematic visualization.

**Table 1 behavsci-14-00252-t001:** Demographics.

		Graduate Students	Non-Student Young Adults
Sample size		*N* = 79	*N* = 59
Age	Range	22–32	22–33
	Average (SD)	25.7 (2.68)	27.3 (3.19)
Sex	Female	93.7% (74 of 79)	94.9% (56 of 59)
	Male	5.1% (4 of 79)	3.4% (2 of 59)
	Other	1.3% (1 of 79)	1.7% (1 of 59)
Education	Some high school	0% (0 of 112)	1.7% (1 of 59)
	High school diploma	0% (0 of 112)	3.4% (2 of 59)
	College degree	68.4% (54 of 79)	52.5% (31 of 59)
	Graduate degree	31.6% (25 of 79)	42.4% (25 of 59)
Racial/Ethnic Identity	American Indian or Alaskan Native	0% (0 of 79)	0% (0 of 59)
	Asian	20.3% (16 of 79)	13.5% (8 of 59)
	Black or African American	6.3% (5 of 79)	5.1% (3 of 59)
	White	64.5% (51 of 79)	72.9% (43 of 59)
	Native Hawaiian or other Pacific Islander	0% (0 of 79)	0% (0 of 59)
	Hispanic, Latino, or Spanish origin	3.8% (3 of 79)	0% (0 of 59)
	Middle Eastern or North African	0% (0 of 79)	1.7% (1 of 59)
	Other	0% (0 of 79)	1.7% (1 of 59)
	Multiracial	3.8% (3 of 79)	5.1% (3 of 59)
	Prefer not to answer	1.3% (1 of 79)	0% (0 of 59)
Previous Care (multiple selections possible)	No previous mental health care	34.2% (27 of 79)	28.8% (17 of 59)
	Outpatient individual therapy	65.8% (52 of 79)	69.5% (41 of 59)
	Outpatient group therapy	10.1% (8 of 79)	8.5% (5 of 59)
	Inpatient care	5.1% (4 of 79)	10.2% (6 of 59)
Incoming Diagnosis	No diagnosis	41.8% (33 of 79)	28.8% (17 of 59)
	Depression	11.4% (9 of 79)	6.8% (4 of 59)
	Bipolar disorder	0% (0 of 79)	0% (0 of 59)
	Anxiety	13.9% (11 of 79)	23.7% (14 of 59)
	Schizophrenia	0% (0 of 79)	0% (0 of 59)
	Personality disorder	0% (0 of 79)	0% (0 of 59)
	PTSD	0% (0 of 79)	0% (0 of 59)
	Other	3.8% (3 of 79)	5.1% (3 of 59)
	Comorbid	29.1% (23 of 79)	35.6% (21 of 59)

**Table 2 behavsci-14-00252-t002:** Quantitative feedback prompts.

Prompt	Average (SD)
How likely are you to recommend Mood Lifters to others? (1–10)	7.9 (2.2)
How useful did you feel Mood Lifters was? (1–10)	7.2 (2.1)
Mood Lifters enables me to better manage my mental health. (Likert, 1–5)	4.1 (0.8)
Because of the work I have done in Mood Lifters, I am performing better at work or in school. (Likert, 1–5)	3.5 (1.0)
I am confident that I could overcome future stressors or mental challenges with the skills I learned in Mood Lifters. (Likert, 1–5)	3.9 (0.8)
I would like to repeat another twelve week session covering these same Mood Lifters modules. (Yes or No)	35.5% Yes
	64.5% No
I would like to be contacted in the future when new Mood Lifters modules are developed. (Yes or No)	80.4% Yes
	19.6% No
Because I’ve got a lot in common with other group members, I’m starting to think that I may have something in common with people outside the group too. (Yes or No)	80.4% Yes
	19.6% No
I would be interested in becoming a Mood Lifters leader. (Yes or No)	30.4% Yes
	69.6% No

## Data Availability

The data that support these findings contain identifiable information and, to protect the confidentiality of participants, have not been made openly available.

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
