# Peer review of "Mood Lifters for Graduate Students and Young Adults: A Mixed-Methods Investigation into Mechanisms of Change in Online Group Therapy"

_behavsci, 2024, doi:10.3390/bs14030252_

Round 1

Reviewer 1 Report

Comments and Suggestions for Authors

The authors develop a study in which they investigate the mechanisms of change induced by the Mood Lifters program in graduate students and young adults through qualitative and quantitative measures. While the study appears to be novel, below, I offer a series of suggestions and recommendations that I believe would enrich it.
The authors invite us to consult detailed information about the ML program's content by directing us to an article currently under review (Pokowitz et al., 2023), which prevents understanding the specific characteristics of this tool. Nevertheless, a more detailed explanation of the Mood Lifters program in the introduction section would be appreciated.
It is necessary to include references that corroborate the effectiveness of the ML program in promoting mental well-being and improving various psychological disorders in the introduction.
A conceptualization of the notion of machine learning is needed.
When the authors discuss research on the mediators and mechanisms underlying the changes observed in various therapeutic modalities, they are referring to studies framed in therapeutic process research that have analyzed those active components in psychotherapy that provide effectiveness to psychological treatments. However, they do not mention at least the main studies in this field. Considering the main objective of this article in terms of investigating the mechanisms of change provided by the ML program, it is especially important to expand the introduction section, including a presentation of the main research that has addressed the issue of factors that make therapies effective. Including references about the study of common factors (of the patient, the therapist, and their interaction) as the main providers of therapeutic effectiveness would enrich the present article.
The authors mention the analysis of qualitative data as a method to explore the mechanisms of change. However, more objective nature evaluations of such change could be added.
The sample sizes are limited (79 graduate students and 59 young adults), which prevents drawing firm conclusions.
In some scientific fields, it may be questionable that peer leaders (students who had previously participated in an ML program) can be assimilated to professional therapists, even though they have received training. It is debatable whether an 8-hour training is sufficient. References supporting such assertions are welcome.
The authors use descriptive statistics as a means to contextualize the qualitative data. I consider that quantitative analyses of this nature do not seem powerful enough to achieve the proposed objectives. An explanation of how the quantitative contextualization of qualitative data can help in detecting and establishing the mechanisms of change through the ML program is also necessary. Arguments based on scientific literature about the value of the mixed methods approach to achieve such an objective are welcome.
Could the fact that the meetings were conducted telematically affect the establishment of a sense of community? Literature on this should be cited.
I understand that the two subject groups (graduate students and young adults) were healthy subjects without diagnosed psychopathology. This means that this study can at most draw conclusions about the improvement of mental well-being in a healthy population but not about the advantages of the program in people suffering from any psychopathology. Additionally, it is not clear whether the ML program acts as a complementary tool to traditional evidence-based therapies or is proposed as an alternative method for the improvement and treatment of psychological imbalances.
In the discussion, it would be useful to connect what is currently known about the factors responsible for therapeutic effectiveness with the benefits provided by the ML program. It would be interesting to question to what extent the benefits provided by the program can be explained from the perspective of therapeutic process research and the importance of common factors in providing improvement to users.
The limitations section should include some of the aspects previously pointed out.

Reviewer 2 Report

Comments and Suggestions for Authors

It is a very interesting manuscript dealing with contemporary and significant issues. I think that the methodology section could be strengthened more if the authors could give a brief description of the assessment inventories used to assess suicidality, mania or psychosis prior to participation to the study. Additionally, authors should mention the exact interval between the end of the program and the post treatment survey and justify the reasons they have not used a follow-up measurement in order to assess the stability/permanence of the potential positive results.  

In regard to limitations, authors should mention that there was no follow-up assessment and that post-treatment survey was based on self-report opinions and emotions. 

I wish you all the best for your current and future research work

Round 2

Reviewer 1 Report

Comments and Suggestions for Authors

The authors have responded in general terms to the proposed revisions and I consider the revision of the manuscript they have submitted to be satisfactory.